

# Exploring Korean adolescent stress on social media: a semantic network analysis

JongHwi Song[1], JunRyul Yang[1], SooYeun Yoo[1], KyungIn Cheon[2], SangKyun Yun[1] and YunHee Shin[2]

[1] Division of Software, Yonsei University, Wonju, Gangwon, Republic of Korea
[2] Wonju College of Nursing, Yonsei University, Wonju, Gangwon, Republic of Korea

## ABSTRACT

**Background.** Considering that adolescents spend considerable time on the Internet and social media and experience high levels of stress, it is difficult to find a study that investigates adolescent stress through a big data-based network analysis of social media. Hence, this study was designed to provide basic data to establish desirable stress coping strategies for adolescents based on a big data-based network analysis of social media for Korean adolescent stress. The purpose of this study was to (1) identify social media words that express stress in adolescents and (2) investigate the associations between those words and their types.

**Methods.** To analyse adolescent stress, we used social media data collected from online news and blog websites and performed semantic network analysis to understand the relationships among keywords extracted in the collected data.

**Results.** The top five words used by Korean adolescents were counselling, school, suicide, depression, and activity in online news, and diet, exercise, eat, health, and obesity in blogs. As the top keywords of the blog are mainly related to diet and obesity, it reflects adolescents' high degree of interest in their bodies; the body is also a primary source of adolescent stress. In addition, blogs contained more content about the causes and symptoms of stress than online news, which focused more on stress resolution and coping. This highlights the trend that social blogging is a new channel for sharing personal information.

**Conclusions.** The results of this study are valuable as they were derived through a social big data analysis of data obtained from online news and blogs, providing a wide range of implications related to adolescent stress. Hence this study can contribute basic data for the stress management of adolescents and their mental health management in the future.

## INTRODUCTION

Desirable mental health is central to adolescent development into healthy adulthood, so multifaceted efforts are urgently needed to properly manage excessive adolescent stress. The 2021 perception rate of stress (38.8%) among Korean adolescents is 32.3% for male

Corresponding author
YunHee Shin, yhshin@yonsei.ac.kr

students and 45.6% for female students, and it tends to increase as they progress to higher grades (*Korea Disease Control and Prevention Agency, 2022*). Additionally, 26.8% of Korean adolescents have experienced depression, 12.7% thought of suicide, and 2.2% have attempted suicide (*Korea Disease Control and Prevention Agency, 2022*). Likewise, approximately 37% of high school students in the United States have experienced periods of persistent feelings of sadness or hopelessness during the past year, and nearly half of all female students in Korea have experienced persistent feelings of sadness or hopelessness in 2019 (*CDC, 2021*).

Adolescents cannot escape from stress in the fierce competition of modern society. Overstressed adolescents often express their stress through delinquency or violence, causing social problems and, in severe cases, leading to suicidal thoughts (*Park, Kang & Lee, 2017*). Adolescents who cannot find an appropriate way to cope with stress levels that exceed their personal resources and can threaten their well-being often attempt suicide as a way of escaping reality (*Kang & Shin, 2015*). Suicidal ideation in adolescents was positively correlated with stress experienced in daily life. Although many previous studies on adolescent stress have been reported, such as factors affecting adolescent stress (*Jang, 2022*), related problems (*Oh & Kweon, 2019*), coping methods, and interventions to reduce stress (*Lim, 2021*), it remains a serious health and social problem.

Korean adolescents spent an average of 285.2 and 397.1 min per day on weekdays and weekends, respectively, on their smartphones (*Korea Disease Control and Prevention Agency, 2022*). Simultaneously, social media use has markedly increased among adolescents. In the US, the proportion of young people between the ages of 13 and 17 years who have a smartphone has reached 89%, more than doubling over a 6-year period, and 70% of teenagers use social media multiple times in a day, up from one-third of all teenagers in 2012 (*Rideout & Robb, 2018*). *Vernon, Modecki & Barber (2018)* analysis of Australian longitudinal data found that 86% of students owned smartphones in Grade 8, increasing to 93% by Grade 11, with increased use of social media communication; additionally, most adolescents rely on smartphones to obtain health information (*Chau, Burgermaster & Mamykina, 2018*).

Recently, with the rapid spread of smartphones, smart TVs, and mobile Internet and social media, the available data has increased exponentially, leading to an era of big data whereby data are used in various fields, particularly in healthcare (*Song, 2013*). Thus, researchers collect social media messages to expand their knowledge and analyse the meaning of the data using social media analytics (*Song & Ryu, 2015*).

Big data in social media is not just a technology to collect, process, and analyse massive amounts of data. The meaning that can be created from such data is more valuable. The core of big data technology is to analyse the pouring information and provide valuable new information and services (*Choi, 2015*). The information on stress on social media can be extremely useful for adolescents, given the amount of time they spend on social media platforms.

Network analysis is a useful way to derive the characteristics of network types and characterise topics of interest in relation to each other (*Kim & Kim, 2016*). Semantic networks were used to infer the subjects used in the texts. Semantic Network Analysis

(SNA) describes the relationships between related concepts through word co-occurrence analysis. By evaluating the networks that appear, the SNA can highlight the most prominent information in the body of the text (*Featherstone et al., 2020*). In addition, such analysis and visualisation help to easily understand the knowledge structure and implicit meaning of the phenomenon of interest (*Yoon, 2013*). Therefore, by analysing and categorising the connectivity of big data-based collection, analysis, and processing, the characteristics and structure of the contents related to stress in adolescents can be identified.

## Previous studies

Previous studies that attempted big data-based SNA on adolescents analysed research trends related to childhood and adolescent cancer survivors in South Korea using word co-occurrence network analysis (*Kang et al., 2021*) and the knowledge structure of students with severe and multiple disabilities (*Song, 2018*). Another study analysed the perception of sports and physical activity in Korean adolescents through big data analysis over the last 10 years, collating data from Naver, Daum, and Google, which are the most widely used search engines (*Park et al., 2020*) in Korea, using TEXTOM 4.0. Yet another study collected data from search engines widely used in Korea to identify social media words that express adolescents' dietary behaviours and identify the associations and types of such words and behaviours. It used text-mining techniques and SNA for related big data collected from the Internet on adolescents' dietary behaviours (*Song et al., 2022*). A study on physical activity and exercise in school-age youth was conducted to provide a solution by analysing a large number of scientific articles through text mining (*Pans et al., 2021*). In the belief that social media plays an important role in adolescents' life, a study describing the big data approach to social media has also been presented by analysing an ad hoc dataset from the eating disorder forum of a social media website (*Moessner et al., 2018*).

Considering the long time spent on internet use and the high levels of stress among adolescents, we had difficulty finding a study that investigated adolescent stress using big data-based network analysis of social media in the process of reviewing previous studies. Therefore, this study was designed to provide basic data to establish desirable stress coping strategies for adolescents, based on big data-based network analysis of social media for Korean adolescent stress. This study aimed to (1) identify social media words that express stress in adolescents and (2) investigate the associations between those words and their types.

## MATERIALS & METHODS

In this study, we used social media big data to analyse adolescents' awareness of stress. The data collection and analysis processes used in this study are shown in Fig. 1 and this is the same method as described in the previous study (*Song et al., 2022*). First, we collected data on adolescent stress by crawling online news and blog websites. Then, we extracted keywords using natural language processing (NLP) and performed pre-processing of the extracted keywords. Next, a SNA was performed to understand the relationships among the extracted keywords. For this study, we implemented two Python programs using suitable libraries instead of using a non-free web-based big data analysis solution such as TEXTOM
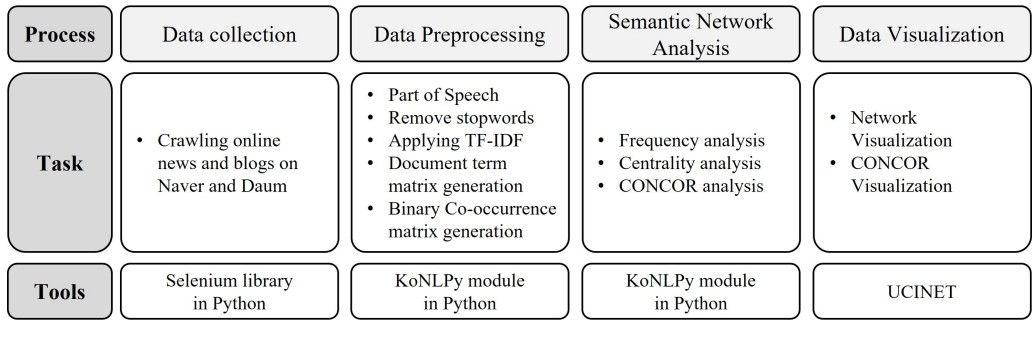

**Figure 1** **The study process.**

(*TheIMC, 2018*); one program collects data by web crawling and the other performs the pre-processing of collected data and SNA, except CONCOR analysis. The UCINET package was used for CONCOR analysis and data visualisation.

## Data collection

We collected relevant data from online news and blogs on *Naver (2022)* and *Daum (2022)*, the largest search engines in Korea. In August 2022, using the search keyword 'adolescent stress', we collected 654 news articles from Naver news and 1,654 blog posts from Naver and Daum blogs using a web crawling programme implemented in Python. Although the period for data collection was not specified, more than 90% of the data are from the last 10 years. We overcame the anti-crawling functions of some websites using the Selenium library, which automates web browser interactions in the programme. Duplicate content frequently occurs in online news because several sources provide news articles with the same. Cosine similarity, defined as the cosine of the angle between two word vectors, is widely used for similarity measurement between documents (*Rahutomo, Kitasuka & Aritsugi, 2012*). We eliminated duplicate content by using cosine similarity.

## Pre-processing

We refined the collected data by selecting only nouns, verbs, and adjectives by the unigram method using KoNLPy, an open-source Python library for Korean NLP (*Park & Cho, 2014*), and excluding stop-words which are commonly used words or unimportant words. Next, we selected the top 30 keywords based on The Term Frequency–Inverse Document Frequency (TF-IDF) values (*Boom et al., 2015*; *Leskovec, Rajaraman & Ullman, 2011*), the opinions of a counselling teacher, and a network analysis expert. The TF-IDF is the formal measure of how the occurrences of a given word are concentrated into relatively few documents. TF-IDF is calculated as $\mathrm{tf}(d,t)\,\mathrm{idf}(d,t)$, where term frequency $\mathrm{tf}(d,t)$ represents the number of appearances of a specific word $t$ in a specific document $d$. Inverse document frequency $\mathrm{idf}(d,t)$ is a factor which diminishes the weight of terms that occur very frequently in the document set and increases the weight of terms that occur rarely; it is represented by $\log(N/(\mathrm{df}(t)+1))$, where $\mathrm{df}(t)$ is the number of documents in which a specific word $t$ appears. Because there is no quantitative criterion for the number of selected keywords,

we have selected top 30 words as in many similar studies (*Choi et al., 2022*; *Jung, 2022*; *Ye et al., 2022*; *Park, Lee & Hong, 2023*).

For the top 30 keywords, we constructed a frequency table; thereafter, a document term matrix (DTM) (*Anandarajan, Hill & Nolan, 2019*) was generated to represent the frequency of words per document. DTM can quantify the relationships between words and documents. Subsequently, a co-occurrence matrix (COM) was constructed to represent the relationship between the simultaneous appearances of words in all documents. To simplify the network analysis of the COM, the constructed COM was transformed into a binary matrix using the median of all its elements as a threshold value. If an element was higher than the threshold value, it was changed to 1; otherwise, it was changed to 0.

## Semantic network analysis and visualisation

SNA was performed to understand the relationships among the top 30 words related to adolescent stress. Network centralities were calculated to identify important words, and a CONvergence analysis of an iterative CORrelation (CONCOR) analysis was performed to identify groups of words with the same relationship pattern.

We calculated four network centralities of COM: (1) degree centrality—the number of nodes a particular node (*Xie, 2005*) is connected to; (2) betweenness centrality—a measure of the mediation role of a node in a network; (3) closeness centrality—the inverse of the mean distance to all other nodes, which indicates how close a node is to all other nodes; and (4) eigenvector centrality—a measure of the influence of a node in a network (*Tabassum et al., 2018*) using the NetworkX (*Hagberg, Swart & Chult, 2008*) Python library.

CONCOR repeatedly partitions nodes into subsets based on structural equivalence and analyses Pearson's correlations to search for groups with certain levels of similarity. It forms clusters, including nodes with similarities to each other (*Breiger, Boorman & Arabie, 1975*). CONCOR analysis was performed using the UCINET 6.0 software package (*Borgatti, Everett & Freeman, 2002*) for the analysis of social networks, and the clustering results were visualised using NetDraw.

# RESULTS

## The frequencies of keywords related to adolescent stress

The frequencies of the top 30 words for adolescent stress in online news and blogs are shown in Table 1. The top five keywords were 'counselling', 'school', 'suicide', 'depression', and 'activity' in online news, and 'diet', 'exercise', 'eat', 'health', and 'obesity' in blogs.

## Analysis of centralities of keywords related to adolescent stress

Table 2 shows the centralities of keywords created using the keyword COM for online news. As the keyword 'counselling' was the most connected with three centralities, it had the highest degree of centrality, followed by 'school', 'suicide', 'problem', 'self-harm', and 'depression'; the highest closeness centrality, followed by 'school', 'suicide', 'problem', 'self-harm', and 'depression'; and the highest eigenvector centrality, followed by 'school', 'suicide', 'problem', 'self-harm', and 'parent'. The keyword 'suicide' had the highest betweenness centrality, followed by 'problem', 'counselling', 'school', 'activity', and 'depression'.

**Table 1  Frequencies of 30 keywords related to adolescent stress in online news and blogs.**

| rank | Word (news) | Freq | Word (blog) | Freq |
|---|---|---|---|---|
| 1 | counseling | 1143 | diet | 6553 |
| 2 | school | 682 | exercise | 3863 |
| 3 | suicide | 654 | eat | 2649 |
| 4 | depression | 629 | health | 2626 |
| 5 | activity | 619 | obesity | 2479 |
| 6 | health | 585 | study | 2007 |
| 7 | education | 509 | counseling | 1659 |
| 8 | problem | 499 | treatment | 1463 |
| 9 | parent | 454 | sleep | 1434 |
| 10 | self-harm | 440 | problem | 1382 |
| 11 | mental health | 423 | parent | 1379 |
| 12 | treatment | 415 | skin | 1354 |
| 13 | family | 403 | weight | 1351 |
| 14 | study | 401 | help | 1316 |
| 15 | experience | 398 | intake | 1241 |
| 16 | participation | 348 | boxing | 1232 |
| 17 | smoking | 335 | acne | 1113 |
| 18 | friend | 331 | friend | 1099 |
| 19 | game | 329 | oneself | 1077 |
| 20 | oneself | 305 | person | 1076 |
| 21 | person | 276 | activity | 1052 |
| 22 | mind | 260 | mind | 915 |
| 23 | relieve | 258 | body | 906 |
| 24 | rest | 242 | rest | 880 |
| 25 | online | 241 | school | 855 |
| 26 | body | 230 | hair loss | 844 |
| 27 | anxiety | 229 | control | 840 |
| 28 | relationship | 226 | worry | 801 |
| 29 | worry | 213 | relieve | 618 |
| 30 | career | 188 | depression | 589 |

Table 3 shows the centralities of keywords which were made by using the keyword COM for blogs. The keyword 'diet' had the highest degree of centrality, followed by 'exercise', 'health', 'study', 'treatment', and 'problem'; the highest closeness centrality, followed by 'exercise', 'health', 'study', 'treatment', and 'problem'. The keyword 'treatment' had the highest betweenness centrality, followed by 'exercise', 'diet', 'study', 'health', and 'sleep'. The keyword 'health' had the highest eigenvector centrality, followed by 'diet', 'exercise', 'problem', 'study', and 'parent'.

## Semantic network of clusters through CONCOR analysis related to adolescent stress

Network groupings and visualisation of adolescent stress are shown in Figs. 2 and 3, respectively. Figure 2 shows the CONCOR analysis of the online news network of adolescent

**Table 2  Centralities of keywords related to adolescent stress from news network.**

| Rank | Keyword | Cd | Keyword | Cb | Keyword | Cc | Keyword | Ce |
|------|---------|-----|---------|-----|---------|-----|---------|-----|
| 1 | counseling | 0.897 | suicide | 0.061 | counseling | 0.906 | counseling | 0.260 |
| 2 | school | 0.897 | problem | 0.061 | school | 0.906 | school | 0.260 |
| 3 | suicide | 0.897 | counseling | 0.058 | suicide | 0.906 | suicide | 0.259 |
| 4 | problem | 0.862 | school | 0.058 | problem | 0.879 | problem | 0.253 |
| 5 | self-harm | 0.828 | activity | 0.043 | self-harm | 0.853 | self-harm | 0.251 |
| 6 | depression | 0.759 | depression | 0.040 | depression | 0.806 | parent | 0.237 |
| 7 | activity | 0.759 | self-harm | 0.036 | activity | 0.806 | family | 0.230 |
| 8 | parent | 0.759 | health | 0.034 | parent | 0.806 | activity | 0.230 |
| 9 | health | 0.724 | parent | 0.030 | health | 0.784 | depression | 0.230 |
| 10 | mental health | 0.724 | mental health | 0.029 | mental health | 0.784 | mental health | 0.228 |
| 11 | family | 0.724 | education | 0.023 | family | 0.784 | health | 0.225 |
| 12 | education | 0.690 | family | 0.022 | education | 0.763 | education | 0.214 |
| 13 | oneself | 0.552 | treatment | 0.011 | oneself | 0.690 | oneself | 0.197 |
| 14 | experience | 0.517 | experience | 0.007 | experience | 0.674 | experience | 0.185 |
| 15 | treatment | 0.483 | oneself | 0.003 | treatment | 0.659 | study | 0.176 |
| 16 | study | 0.483 | study | 0.003 | study | 0.659 | friend | 0.174 |
| 17 | friend | 0.483 | friend | 0.003 | friend | 0.659 | treatment | 0.161 |
| 18 | person | 0.414 | person | 0.000 | person | 0.630 | relationship | 0.156 |
| 19 | relationship | 0.414 | relationship | 0.000 | relationship | 0.630 | person | 0.154 |
| 20 | mind | 0.310 | body | 0.000 | mind | 0.592 | mind | 0.121 |
| 21 | anxiety | 0.310 | game | 0.000 | anxiety | 0.592 | anxiety | 0.118 |
| 22 | participation | 0.276 | participation | 0.000 | participation | 0.580 | relieve | 0.110 |
| 23 | relieve | 0.276 | smoking | 0.000 | relieve | 0.580 | participation | 0.108 |
| 24 | game | 0.241 | mind | 0.000 | game | 0.569 | online | 0.096 |
| 25 | rest | 0.241 | relieve | 0.000 | rest | 0.569 | rest | 0.094 |
| 26 | online | 0.241 | rest | 0.000 | online | 0.558 | career | 0.092 |
| 27 | career | 0.241 | online | 0.000 | career | 0.558 | game | 0.090 |
| 28 | smoking | 0.172 | anxiety | 0.000 | body | 0.537 | worry | 0.071 |
| 29 | body | 0.172 | worry | 0.000 | worry | 0.537 | smoking | 0.063 |
| 30 | worry | 0.172 | career | 0.000 | smoking | 0.527 | body | 0.060 |

**Notes.**

Cd, Degree Centrality; Cb, Betweenness Centrality; Cc, Closeness Centrality; Ce, Eigenvector Centrality.

stress consisting of five clusters. We represented the cluster consisting of words 1, 2, …
in [word 1, word 2, …]. The cluster [body, smoking, person, anxiety, mind, treatment,
rest] could be considered 'a pattern that occurs when adolescents are under stress', as the
cluster reflects having an anxious mind, searching for someone to be with, smoking, resting
one's body, or receiving treatment. The cluster [problem, parent, health, mental health,
depression, experience, oneself] can be interpreted as 'the causes and consequences of
adolescent stress', as adolescents' experiences with problems between themselves and their
parents affect their health and mental health, especially leading to depression. The cluster
[worry, game, activity, relationship, friend] could be regarded as 'a way for adolescents
to relieve stress', that is, to find a friend to relieve stress, talk about their worry, and play

**Table 3  Centralities of keywords related to adolescent stress from blog network.**

| Rank | Keyword | Cd | Keyword | Cb | Keyword | Cc | Keyword | Ce |
|---|---|---|---|---|---|---|---|---|
| 1 | diet | 0.862 | treatment | 0.068 | diet | 0.879 | health | 0.254 |
| 2 | exercise | 0.862 | exercise | 0.068 | exercise | 0.879 | diet | 0.254 |
| 3 | health | 0.862 | diet | 0.061 | health | 0.879 | exercise | 0.253 |
| 4 | study | 0.828 | study | 0.058 | study | 0.853 | problem | 0.247 |
| 5 | treatment | 0.793 | health | 0.057 | treatment | 0.829 | study | 0.245 |
| 6 | problem | 0.793 | sleep | 0.034 | problem | 0.829 | parent | 0.240 |
| 7 | counseling | 0.759 | counseling | 0.032 | counseling | 0.806 | counseling | 0.234 |
| 8 | parent | 0.759 | help | 0.027 | parent | 0.806 | eat | 0.234 |
| 9 | eat | 0.724 | problem | 0.026 | eat | 0.784 | treatment | 0.233 |
| 10 | sleep | 0.690 | parent | 0.021 | sleep | 0.763 | obesity | 0.216 |
| 11 | obesity | 0.655 | eat | 0.015 | obesity | 0.744 | sleep | 0.215 |
| 12 | help | 0.655 | friend | 0.014 | help | 0.744 | help | 0.209 |
| 13 | oneself | 0.586 | activity | 0.012 | oneself | 0.707 | person | 0.204 |
| 14 | person | 0.586 | obesity | 0.011 | person | 0.707 | oneself | 0.201 |
| 15 | activity | 0.586 | oneself | 0.007 | activity | 0.707 | activity | 0.198 |
| 16 | friend | 0.552 | person | 0.006 | friend | 0.690 | friend | 0.186 |
| 17 | mind | 0.448 | skin | 0.005 | mind | 0.644 | mind | 0.165 |
| 18 | body | 0.448 | body | 0.003 | body | 0.644 | body | 0.157 |
| 19 | weight | 0.414 | worry | 0.002 | weight | 0.630 | school | 0.143 |
| 20 | school | 0.379 | weight | 0.002 | school | 0.617 | control | 0.142 |
| 21 | control | 0.379 | control | 0.001 | control | 0.617 | weight | 0.140 |
| 22 | rest | 0.345 | intake | 0.000 | rest | 0.604 | rest | 0.133 |
| 23 | intake | 0.310 | mind | 0.000 | intake | 0.580 | intake | 0.107 |
| 24 | worry | 0.276 | school | 0.000 | worry | 0.580 | worry | 0.100 |
| 25 | skin | 0.241 | boxing | 0.000 | skin | 0.558 | depression | 0.089 |
| 26 | depression | 0.241 | acne | 0.000 | depression | 0.547 | skin | 0.073 |
| 27 | acne | 0.138 | rest | 0.000 | acne | 0.518 | relieve | 0.052 |
| 28 | relieve | 0.138 | hair loss | 0.000 | hair loss | 0.518 | hair loss | 0.041 |
| 29 | boxing | 0.103 | relieve | 0.000 | relieve | 0.518 | acne | 0.041 |
| 30 | hair loss | 0.103 | depression | 0.000 | boxing | 0.500 | boxing | 0.039 |

**Notes.**

Cd, Degree Centrality; Cb, Betweenness Centrality; Cc, Closeness Centrality; Ce, Eigenvector Centrality.

games or activities. The cluster [relieve, study, online, career, participation] could be interpreted as 'to relieve the stress of studying, they participate in events such as online career experiences'. The cluster [suicide, self-harm, school, family, education, counselling] could be considered 'education and counselling about self-harm or suicide due to stress is required at school and in the family'.

Figure 3 shows the CONCOR analysis of the blog network of adolescent stress, comprising seven clusters. The clusters [acne, skin, weight, intake] and [eat, obesity, health, diet, exercise, problem, parent] can be considered 'sources of stress'; the cluster [school, rest, friend, boxing, relief] as in 'coping with stress such as spending time with friends at school, resting, or boxing'; the cluster [study, counselling, activity] could be interpreted as 'coping

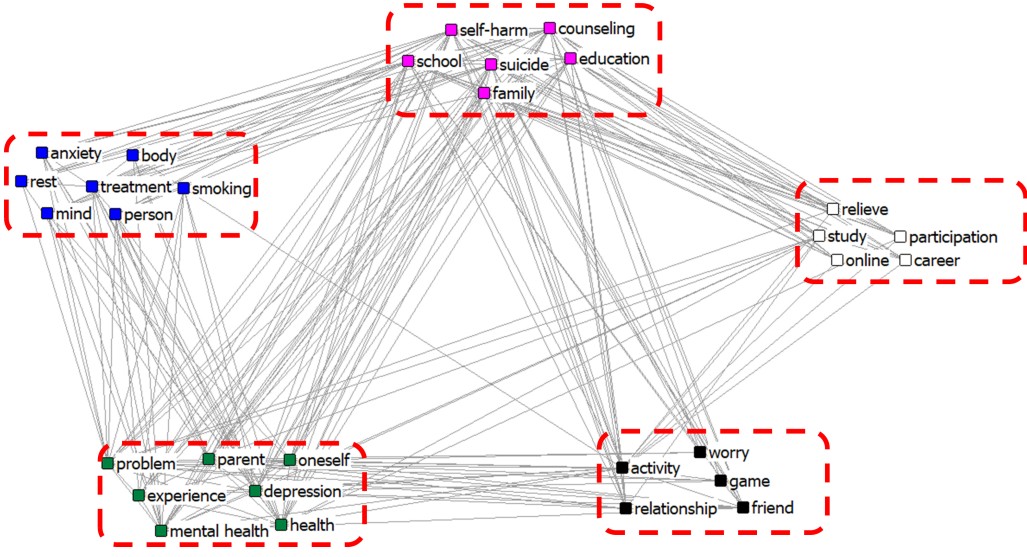

**Figure 2   CONCOR analysis of news network of the adolescent stress.**

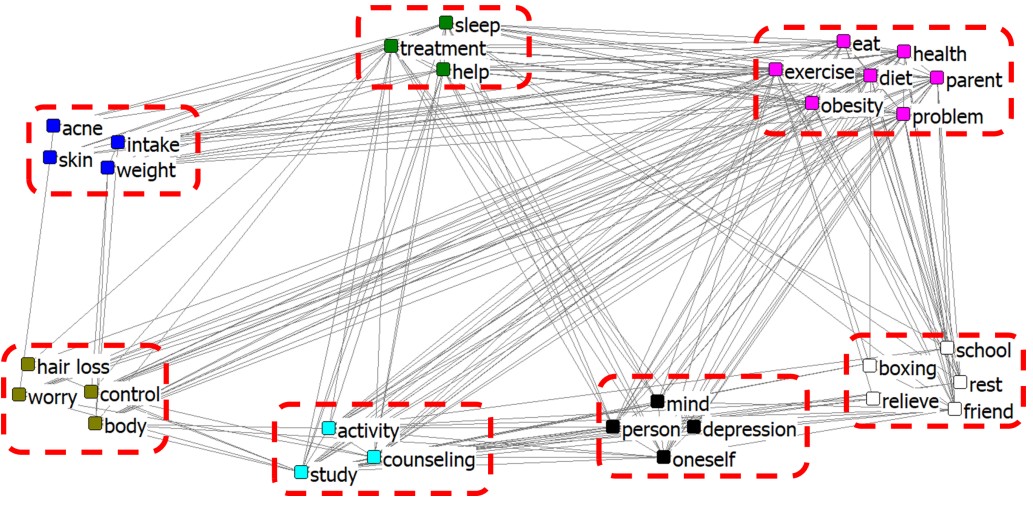

**Figure 3   CONCOR analysis of blog network of the adolescent stress.**

with the stress by studying, counselling, or activities'. The cluster [person, mind, depression, oneself] could be interpreted as 'relationships between themselves and others causing stress and depression', and the cluster [hair loss, control, body, worry] could be considered 'a phenomenon that can occur when adolescents are under stress' which were reflected in hair loss, (emotional) control, body (imbalance), and worry. The cluster [treatment, help, sleep] could be seen as 'strategies adolescents can adopt to relieve stress'.

## DISCUSSION

Since the mental health of adolescents is extremely important for their future growth into healthy adults, various efforts are urgently needed to appropriately manage their current levels of stress. Therefore, in this study, we analysed online social big data on adolescent stress using text mining techniques.

As a result of text mining of keywords related to adolescent stress, the top five words with high frequency were 'counselling', 'school', 'suicide', 'depression', and 'activity' in online news, and 'diet', 'exercise', 'eat', 'health', and 'obesity' in blogs. The tendency of the top five words appeared such that the words of the blogs appeared somewhat lighter than those of the online news; this reflects the fact that the blogs were lighter, more personal, and informal, and these features distinguished them from the news (*Tereszkiewicz, 2014*). The fact that the top keywords of the blog were mainly related to diet and obesity reflects the high interest of adolescents in their bodies (*Yun, 2018*), which was also confirmed as a source of immense stress among adolescents.

Analysis of online news revealed that words that refer to resolving stress or behaviours caused by stress—such as counselling, school, self-harm, problem, and suicide—have high centrality. In contrast, in the centrality analysis of blogs, although treatment was highlighted, high centrality words reflecting the cause of stress—such as diet, study, and health—were also indicated. The result of 'counselling' having the highest connection with other keywords in network centrality analysis of the online news may indicate a lot of resolution in consideration of the social issues of adolescent stress due to the nature of news which has a formal character (*Jeong & Kim, 2010*). As the characteristics of blogs are considered temporary, personal, and informal (*Thorsen & Jackson, 2018*), it was confirmed that adolescents frequently shared personal causes of stress through individual blogs. Furthermore, since social media can influence adolescents' self-views and interpersonal relationships through social comparisons and negative interactions, social media content often promotes self-harm and suicidal thoughts among adolescents (*Abi-Jaoude, Naylor & Pignatiello, 2020*). Corroborating the descriptions in this last cited study, another study reported that adolescents with depression and/or suicidality often use more social media and report worsening mood and suicide risk (*John et al., 2018*). Nonetheless, researchers also found that lower levels of social media use (overall and messaging) are associated with a greater likelihood of having suicidal ideation with plan over the next 30 days (*Hamilton et al., 2021*). Therefore, it is relevant to consider the importance of social media as an additional context for the topic of adolescent suicide and to educate adolescents to avoid placing indiscriminate trust on social media.

Through CONCOR analysis, five clusters were identified in online news and seven in blogs. The causes and symptoms of stress and coping strategies were confirmed in both online news and blogs. However, online news contained multiple coping strategies for relieving stress, whereas blogs focused more on the causes of stress. 'Study' as a cause of stress was included in both online news and blogs. Diet, obesity, acne, skin, hair loss, and various other causes of stress, such as school, and family, have been included in blogs and highlighted in previous studies. Contrarily, online news focused more on coping strategies

for relieving adolescent stress. The importance of counselling for adolescent concerns, education, and counselling in schools and families was also confirmed. Korean adolescents are under a lot of stress due to their studies, and with excessive academic stress, they may experience mental health-related problems, such as depression (*Kang, 2022*). Because modern society tends to judge and evaluate people based on their appearance, it is common for adolescents to rate themselves based on their appearance. Acne occurrence is associated with stress and depression; therefore, acne treatment and skin care are considered necessary for improving mental health (*Shin & Kim, 2019*).

Our analyses indicate that blogs contain more content about the causes and symptoms of stress than online news, reflecting the trend of social networks of casual and informal youth blogging as a new channel for sharing personal information (*Li et al., 2016*). In one study, researchers designed and implemented a microblogging platform to detect and relieve stress in teenagers; the authors mentioned the potential for stress detection because stressed individuals view microblogs as a channel for emotional release and interaction (*Zhao et al., 2016*). Young people who experience illnesses, including stress, tend to blog about them, and such blogs often have many followers. By means of blogging, young people living with an illness may succeed in having a social life and uphold and even extend their self-knowledge and self-esteem. The content of a blog can foster familiarity between the author and readers. Blogs can provide the authors' unique experience-based knowledge and reflection to readers who read published articles. Therefore, blogging, especially on specific issues involving stress, should be continuously explored and recognised as a valuable source for such content in the future (*Nesby & Salamonsen, 2016*). This finding suggests that differences in blog authors' subjective thoughts and direct experiences are used as the main basis for blogs, whereas online news focuses on delivering objective information and explanations based on the values of fairness and responsibility (*Jeong & Kim, 2010*). Nowadays, Internet usage is unavoidable for the younger generations. The online world is the primary source of information and quick communication; therefore, education about the correct use of the internet should be made reasonable at the earliest (*Prievara & Piko, 2016*).

Finally, the results suggest that when establishing a stress coping strategy for adolescents, first, the information currently present in social media can be utilised by stakeholders; this is in consideration that blogs focused more on the causes of stress and online news contain relatively large amounts of information on stress coping. Second, since elements related to appearance and academics are often cited as causes of stress for Korean adolescents, they should be incorporated into stress coping interventions aimed at this population. Third, social media could be given greater importance within the context of adolescent suicide.

## CONCLUSIONS

To contribute to a strategy for preventing and managing adolescent stress, we analysed social media data using text-mining techniques and derived the words and word associations from online news and blog content. We collected data from the two largest portals for Koreans, including Korean adolescents, using the search term 'Adolescent stress' and

related words. In this study, we collected only data that is in Korean from Korean portal sites. Although information about adolescent stress is available on websites worldwide, only data in Korean language were collected to ensure consistency with keyword selection. Despite these limitations, the results of this study are valuable as they were derived through social big data analysis of data obtained from online news and blogs. The findings provide a wide range of implications related to adolescent stress; hence this study can contribute as basic data for the stress management of adolescents and their mental health management in the future.

### Funding
This work was supported by the National Research Foundation of Korea (NRF) grant funded by the Korean government [Ministry of Science and ICT (MSIT)] (No. 2021R1F1A106253211). The funders had no role in study design, data collection and analysis, decision to publish, or preparation of the manuscript.

### Grant Disclosures
The following grant information was disclosed by the authors:
National Research Foundation of Korea (NRF): 2021R1F1A106253211.

### Competing Interests
The authors declare there are no competing interests.

### Author Contributions
- JongHwi Song conceived and designed the experiments, performed the experiments, analyzed the data, prepared figures and/or tables, authored or reviewed drafts of the article, and approved the final draft.
- JunRyul Yang conceived and designed the experiments, performed the experiments, analyzed the data, prepared figures and/or tables, authored or reviewed drafts of the article, and approved the final draft.
- SooYeun Yoo performed the experiments, prepared figures and/or tables, and approved the final draft.
- KyungIn Cheon analyzed the data, authored or reviewed drafts of the article, and approved the final draft.
- SangKyun Yun conceived and designed the experiments, analyzed the data, prepared figures and/or tables, authored or reviewed drafts of the article, and approved the final draft.
- YunHee Shin conceived and designed the experiments, analyzed the data, prepared figures and/or tables, authored or reviewed drafts of the article, and approved the final draft.

### Data Availability
The data is available at Zenodo: Shin, YunHee, Yun, SangKyun, Song, JongHwi, Yang, JunRyul, Yoo, SooYeun, & Cheon, KyungIn. (2023). Exploring Korean adolescent stress on social media: A semantic network analysis [Data set]. Zenodo. https://doi.org/10.5281/zenodo.7602111.

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
