# Peer review of "Exploring Korean adolescent stress on social media: a semantic network analysis"

_PeerJ, doi:10.7717/peerj.15076_

## Round 0.1 · original submission · Major Revisions

Your manuscript has been reviewed and assessed by two reviewers, and both of them agree with the fact that there are still a few points that need to be addressed. The comments of the reviewers are included at the bottom of this letter. Reviewers indicated that the introduction, methods, and results sections should be improved. We would be glad to consider a substantial revision of your work, where the reviewer’s comments will be carefully addressed one by one.

Additionally, I also have recommendations below:
• Information about the time period in which the data was collected should be shared in the method section.
• The names of all programs used in the analysis should be included.
• Provide the names of the list for online news and blog websites.

Reviewer 1 ·

Basic reporting

See my comments below

Experimental design

See my comments below

Validity of the findings

See my comments below

Additional comments

• P8~L137-144: The authors mentioned “…collected data by selecting only nouns, verbs, and adjectives…” and nouns in Tables 1-3 on P17-21 (such as mental health, school,…). For word extraction, which one (unigram, bi-gram,…) did the authors use?
• P8~L148-151: The authors mentioned “For the top 30 words, a document term matrix (DTM) (Anandarajan, Hill & Nolan, 2019) was generated to represent the frequency of words per document. DTM can quantify the relationships between words and documents. Subsequently, a co-occurrence matrix (COM) was constructed to represent the relationship between the simultaneous appearances of words in all documents…”. How did the authors select words depicting “stress” from the list of extracted words? By manual selection from experts or automatic selection? If the automatic selection, how did the authors do?
• P8~L148-151: The authors mentioned “For the top 30 words…”. Why did the authors select the top 30 words? Could you explain more or any reason(s)?
• The results (e.g., visualization, etc.): There are no evaluation or comparison to the other studies or tools/software about the top 30 words and network in order to assess the proposed study process being better than the others.
• P22~Figure 1: The phases of the proposed study process used tools/library (e.g., KoNLPy, etc.) What is the major contribution of the study? What is difference between the proposed study process and the other?

Reviewer 2 ·

Basic reporting

In this work, the authors analyze social media data to identify terms related to adolescent stress in Korea. For this study, they collected data from Korean news and blog portals. Their findings include possible main reasons causing stress and the ways adolescents express their concerns. The work is well written and presented. The analysis of the results is clear and appropriate. I like the visualization of samples in the dataset and analysis of the results, where we can appreciate the groups of themes related to stress and mental health.

I have some comments that could help to improve the paper:
The pages in my pdf file are not numbered, so I will count it as page 1 where the line count starts.

page 1:
- Background. Considering that adolescents spend considerable time on the Internet and social media and experience high levels of stress. -> It seems that in this sentence the reason for the stress is the time spent on the internet and social networks. Is that so or did you mean something else?
- The purpose of this study was to identify (1) social media words -> (1) identify social media words.

page 2:
- Korean adolescents are 32.3% for male students and 45.6% for female students, and it tends to increase as they progress to higher grades. -> is a very interesting fact. Did you find in your analysis any relation to these results? For example, what is mentioned below related to weight and appearance within the concerns.
- A couple of times US statistics are mentioned and it makes me think that they are going to make a comparison with them directly but it is not the case.
- Something that seems strange to me is that the introduction and the related work are in one section. I would recommend separating them. In this way, in the introduction part, you can add more clearly the motivations, contributions, and other relevant information that will be contemplated in the article. Also, by creating a related work section, it would be more explicit for the reader where your work is in comparison to what already exists.

page 4:
Data collection.
- We eliminated duplicate content by using cosine similarity -> explain in more detail how you do this part, and in general, this subsection needs more information.

page 7:
- social media content often promotes self-harm and suicidal thoughts among adolescents -> this is a strong claim, you need to add examples or more citations that support it.

General comments:
- I am not sure why the tables and figures were added at the end of the paper. This was requested by the format of the journal. If not, you should place them as close as possible to where the text mentions them, making it easier for the reader to see what is being explained about them.
- Figures 2 and 3 could have a smaller space between the clusters and thus look better.

Experimental design

As mentioned in my previous comments, I wish the authors could expand the data collection section a bit more or add general statistics of the dataset and move Figure 1 where this process can be better seen.

Validity of the findings

It seems to me that the topic addressed is of great current importance due to the great problems that can arise from the stressful environment that adolescents are subjected to. The authors mention that with their analysis they could create strategies to improve this situation, but they have not added examples, I wish they could expand a little more on this part. I would like them to connect the information they found with some strategy or recommendation they would make based on it.

---

## Round 0.2 · Minor Revisions

There are still a few issues that need to be addressed. Please respond to the Reviewer 1’s comments seriously. I invite you to respond to the comments and revise your manuscript.

Reviewer 1 ·

Basic reporting

See my comments below

Experimental design

See my comments below

Validity of the findings

See my comments below

Additional comments

a. There are some points replied and mentioned by the authors. However, could the authors explain or point out to make your study being better?

• P8~L137-144: The authors mentioned “…collected data by selecting only nouns, verbs, and adjectives…” and nouns in Tables 1-3 on P17-21 (such as mental health, school,…). For word extraction, which one (unigram, bi-gram,…) did the authors use?
 The authors replied: P8~L144: …. by the unigram…
 What is difference between “mental health” and “health” in Korean language? Those are extracted with unigram method or not?
• P8~L148-151: The authors mentioned “For the top 30 words…”. Why did the authors select the top 30 words? Could you explain more or any reason(s)?
 There is no mention about this point.
• The results (e.g., visualization, etc.): There are no evaluation or comparison to the other studies or tools/software about the top 30 words and network in order to assess the proposed study process being better than the others.
 There is no mention about this point.
• P22~Figure 1: The phases of the proposed study process used tools/library (e.g., KoNLPy, etc.) What is the major contribution of the study? What is difference between the proposed study process and the other?
 The authors mentioned:
+ P8~120-121: … and this is the same method as described in the previous study (Song et al., 2022)…
+ P7~114-115: … This study aimed to (1) identify social media words that express stress in adolescents and 115 (2) investigate the associations between those words and their types…
 All of phases in Figure 1 for the study process diagram are the exact same, except arrows being connected between phases in the other publication. The diagram had been found by searching from the internet in the previous review. Hence, the point “What is difference between the proposed study process and the other?” would need to be concerned by the authors. It is related to plagiarism. Assume: the publication published online, and this manuscript are the same authors. It is also considered as self-plagiarism.
b. Some points would be more details by the authors.
• The results (keywords) depicted on P24-25 (Figures 2-3) are in English. Did the authors analyze on Korean language, and then represent in English after translating or not?
• To represent data visualization, the authors mentioned the tool UCINET on P23 (Figure 1). The UCINET is a commercial software. To visualize the clustering result, the authors mentioned the tool NetDraw on P9 (Lines 180-181). The UCINET (NetDraw) is a commercial software. The authors should mention the license.

Reviewer 2 ·

Basic reporting

I thank the authors for taking into account our comments to improve the paper. I consider that the article is ready to be published in its current state.

Experimental design

Both the methodology and the analysis within the study are sufficient for publication.

Validity of the findings

The topic addressed is of current importance due to the problems that can arise from the stressful environment that adolescents are subjected to.

---

## Round 0.3 · accepted · Accept

The authors addressed the reviewers' concerns and substantially improved the content of the manuscript. So, based on my assessment as an academic editor, the manuscript can be accepted in its current form.

Reviewer 1 ·

Basic reporting

All are clear and good explanation.

Experimental design

The study's methodology and analysis are satisfactory for publication.

Validity of the findings

The subject being discussed is relevant because adolescents are exposed to a stressful environment that can lead to various issues.

Additional comments

None